# WEAKLY-SUPERVISED CAMERA LOCALIZATION BY GROUND-TO-SATELLITE IMAGE REGISTRATION

## ABSTRACT

The ground-to-satellite image matching/retrieval was initially proposed for city-scale ground camera localization. Recently, more and more attention has been paid to increasing the camera pose accuracy by ground-to-satellite image matching, once a coarse location and orientation has been obtained from the city-scale retrieval. This paper addresses the same scenario. However, existing learning-based methods for solving this task require accurate GPS labels of ground images for network training. Unfortunately, obtaining such accurate GPS labels is not always possible, often requiring an expensive Real Time Kinematics (RTK) setup and suffering from signal occlusion, multi-path signal disruptions, *etc*. To address this issue, this paper proposes a weakly supervised learning strategy for ground-to-satellite image registration when only noisy pose labels for ground images are available for network training. It derives positive and negative satellite images for each ground image and leverages contrastive learning to learn feature representations for ground and satellite images useful for translation estimation. We also propose a pseudo image pair creation strategy for cross-view rotation estimation network training. Experimental results show that our weakly-supervised learning strategy achieves the best performance on cross-area evaluation, compared to the recent state-of-the-art methods that require accurate pose labels for supervision, and shows comparable performance on same-area evaluation.

## 1 INTRODUCTION

Camera localization is pivotal in real-world applications such as autonomous driving, field robotics, and augmented/virtual reality. Recent research has explored diverse methods to approximate the coarse location and orientation of a ground camera. These methods encompass noisy sensors like consumer-level GPS and compass, as well as visual retrieval techniques, *et al*. To attain greater pose accuracy, other sensors like Lidar (Vora et al., 2020; Mishra et al., 2022; De Paula Veronese et al., 2015), Radar (Tang et al., 2020; 2021), and High Definition (HD) maps, have been investigated. However, many commercial autonomous vehicles at level two/three lack these sensors. Maintaining and updating high-precision HD maps is challenging and expensive. In response, satellite imagery has emerged as a viable alternative reference source due to its wide accessibility and global coverage.

We focus on ground-to-satellite camera localization, aiming to determine a ground camera's location and orientation relative to a geo-tagged satellite map. Prior research (Workman & Jacobs, 2015; Vo & Hays, 2016; Hu et al., 2018; Liu & Li, 2019; Zhai et al., 2017; Regmi & Shah, 2019; Shi et al., 2019; Toker et al., 2021; Zhu et al., 2022) has centered on city-scale camera localization, employing image retrieval to match ground and satellite images. However, image retrieval introduces errors that can span tens of meters. Recent efforts have addressed this by enhancing camera pose accuracy through ground-to-satellite image registration, guided by coarse location and orientation estimates. Nonetheless, the significant viewpoint differences between ground and satellite images make handcrafted features fail (Shi et al., 2022). Learning-based approaches (Zhu et al., 2021b; Xia et al., 2022; Shi & Li, 2022; Lentsch et al., 2023; Fervers et al., 2022) require a large training dataset with accurate GT poses for the ground images, which can be laborious and expensive to obtain, and even the high-accuracy RTK GPS can suffer from inaccurate locations when correction signals are absent (Geiger et al., 2013; Maddern et al., 2017). Therefore, this paper aims to develop a weakly supervised ground-to-satellite image registration strategy to increase the ground cameras' pose accuracy when only coarse pose labels for ground images are provided.

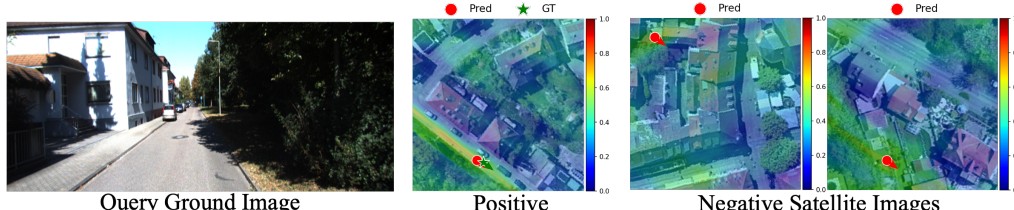

Figure 1: We derive positive and negative satellite images for each ground-view image based on its coarse location information. A similarity map between the ground view and each satellite image is computed when aligned at different locations. Our training objective aims to maximize the maximum similarity in the positive similarity map while minimizing the maximum similarity in the negative similarity map.

We target the 3-DoF (degree-of-freedom) pose estimation for ground cameras, *i.e.*, 2-DoF location and 1-DoF orientation. Under the weakly-supervised application scenario, deterministic pose output by a network is suboptimal, as no GT pose is available to supervise the network output. We resort to the recent representation learning approaches by contrastive learning to solve this problem.

Given the coarse pose of a ground camera, we determine a satellite image that covers the local surroundings of the camera and some satellite images that do not cover the local surroundings, which are regarded as positive and negative for this ground image, respectively. We utilize the signal that a ground image is within its positive satellite image while outside its negative satellite images to train a network. The network is trained to learn feature representations such that the similarity between the ground and its positive satellite images at their optimal relative pose is more significant than between the ground and its negative satellite images at their "optimal" relative pose.

The "optimal" relative pose for a ground-and-satellite image pair is determined by the maximum cross-view similarity when aligned at this relative pose (among others). We follow the standard synthesis-and-matching procedure to determine this "optimal" relative pose: first synthesizing an overhead-view feature map[1] from the ground view image and then matching it against the reference satellite feature map. Conventional methods for this matching process rotate and translate the synthesized overhead view feature map according to predetermined candidate poses. Considering the vast search space for the 3-DoF pose, we decouple the rotation and translation estimation.

The rotation estimation is conducted first, and the framework is designed by network regression. It takes input as the query and reference images and outputs their relative pose. To supervise this network, we create some "satellite and satellite" image pairs with GT relative poses. Specifically, we randomly rotate and translate a satellite image. The transformed satellite image mimics a synthesized overhead view image from a ground image. We train the network to estimate the relative pose between the original and transformed satellite images. Once the network is trained, it is leveraged to estimate the relative orientation between a query ground image and its positive satellite image.

After this, our translation estimation framework synthesizes an overhead-view feature map from the ground image with the orientation aligned with satellite images and then matches it against a satellite feature map. The output is a similarity (/location probability) map of the ground image with respect to the satellite image. This process is implemented between a query image and its positive and negative satellite images, as illustrated in Fig. 1. The contrastive learning supervision strategy illustrated above maximizes the maximum similarity in the positive similarity map while minimizing the maximum similarity in the negative similarity map.

We conduct experiments on a popular autonomous driving dataset where ground images are captured by a pin-hole camera with limited field-of-view (FoV), KITTI (Geiger et al., 2013), and a well-known cross-view localization dataset where ground images are panoramas, VIGOR (Zhu et al., 2021b). Experimental results demonstrate that our method achieves the best generalization ability compared to the recent state-of-the-art despite not requiring accurate pose labels for supervision.

## 2    RELATED WORK

We discuss related works on ground-to-satellite localization and self/weakly supervised learning.

---

[1]Satellite images used for ground camera localization are usually roughly orthographic.

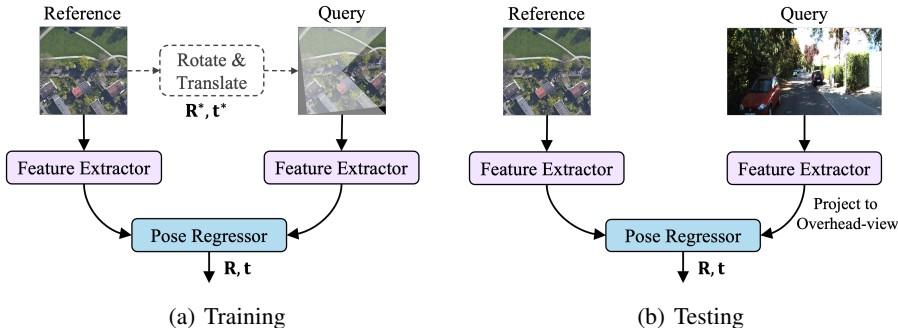

|  (a) Training | (b) Testing |
| --- | --- |

Figure 2: Self-supervised rotation estimator.

## 2.1 GROUND-TO-SATELLITE IMAGE-BASED LOCALIZATION

**City-scale localization.** Ground-to-satellite image-based localization aims to determine the location of a ground camera by matching it with a satellite map covering the region of interest. Initially proposed for city-scale coarse-level localization, it was formulated as an image retrieval problem. Specifically, a large satellite map of the interested region is first split into small patches to construct a geo-referenced satellite image database. For a query image captured at the ground level, its similarity with every database satellite image is computed, and the GPS of the most similar satellite image is taken as the query camera's location. Over the past decades, hand-crafted features (Castaldo et al., 2015; Lin et al., 2013; Mousavian & Kosecka, 2016) have been demonstrated to be a bottleneck for cross-view feature matching due to significant geometric and appearance variations. Seminal works using deep networks (Workman & Jacobs, 2015; Workman et al., 2015; Vo & Hays, 2016) demonstrated that the learned feature descriptors by a metric learning training objective offer better and more reliable performance. Researchers have since investigated learning powerful and discriminative feature descriptors (Cai et al., 2019; Yang et al., 2021; Zhu et al., 2022), orientation-aware cross-view representations (Hu et al., 2018; Liu & Li, 2019; Sun et al., 2019; Zhu et al., 2021a), and different strategies for bridging the cross-view domain gap (Zhai et al., 2017; Regmi & Shah, 2019; Shi et al., 2019; 2020b; Toker et al., 2021). Cross-view localization has also been extended from a 2-DoF location estimation task to a 3-DoF joint location and orientation estimation task (Shi et al., 2020a), and from a single image-based localization problem to a video-based localization problem (Vyas et al., 2022; Shi et al., 2023b; Zhang et al., 2023). However, the database images are often discretely sampled, while query images' locations are in a continuous region. Thus, the image retrieval formulation only results in a coarse camera pose estimation, and its accuracy is determined by the sample density of ground images.

**Increasing localization accuracy.** Recently, researchers have started to investigate how to increase the accuracy of a ground camera's pose by ground-to-satellite image matching once the camera's coarse rotation and orientation have been determined. Towards this purpose, network regression (Zhu et al., 2021b), pose optimization (Shi & Li, 2022) and similarity matching based methods (Xia et al., 2022; Lentsch et al., 2023; Xia et al., 2023; Fervers et al., 2022; Shi et al., 2023a; 2022; Sarlin et al., 2023) have been explored. Nonetheless, all these works need sub-meter and sub-degree pose labels for ground images in the training data to train their networks. In this paper, we propose a strategy that estimates the relative rotation and translation between a ground and a satellite image when such accurate labels are unavailable in the training data.

## 2.2 SELF/WEAKLY-SUPERVISED LEARNING

Self- and weakly-supervised learning has been widely explored in other tasks, such as image classification (Zhai et al., 2019), object detection (Dang et al., 2023), semantic segmentation (Wang et al., 2022), image inpainting (Pathak et al., 2016), point cloud registration (Liu et al., 2023), human/hand/object pose estimation (Bouazizi et al., 2021; Spurr et al., 2021; Gharaee et al., 2023), the intersection between vision and natural language processing, (Radford et al., 2021) *et al*. Many of them also exploit contrastive learning as supervision signals. Tang *et al*. (Tang et al., 2021) proposed a self-supervised strategy for localizing outdoor range sensors (*e.g*., Lidar, Radar). However,

little attention has been paid to camera pose estimation with self or weak supervision. In this work, we introduce the first weakly-supervised learning strategy for ground camera pose refinement by ground-and-satellite image registration. The technical details are illustrated next.

## 3 METHOD

Given a coarse location and orientation of a ground camera, our goal is to refine this pose through ground-to-satellite image registration. In contrast to previous works that assume a large training dataset with highly accurate pose labels for ground images, we propose a weakly supervised learning strategy that does not require such labels. Our approach first estimates the rotation of the ground camera by network regression and then computes the translation by similarity matching.

### 3.1 ROTATION ESTIMATION BY SELF-SUPERVISION

We draw inspiration from Spatial Transformer Networks (Jaderberg et al., 2015) to regress relative orientations between cross-view images by a network. The network takes the ground (query) and satellite (reference) images as input and outputs their relative pose, as illustrated in Fig. 2(b). During inference, the network first extracts feature representations from the satellite and ground images separately. Then, the ground features are projected to the overhead view according to the ground plane Homoraphy, as in Shi & Li (2022). Next, a Pose Regressor, constructed by neural networks, is employed to estimate the relative pose between the projected overhead view and the reference satellite feature map.

To supervise the network, we generate some "satellite-and-satellite" image pairs with GT relative poses. This is done by rotating and translating a satellite image using a randomly generated pose, $\mathbf{R}^*, \mathbf{t}^*$, where the maximum magnitude of the rotation angle and translation is based on the error of the ground camera's coarse pose information that we aim to refine during deployment. Furthermore, we apply a mask on the transformed satellite image and extract a triangle region corresponding to the ground camera's FoV, as shown at the top of Fig. 2(a). The triangle region corresponds to the ground camera's horizontal field of view, computed from its focal length and image size. This is to mimic a synthesized overhead view image from a ground-view image. We refer to the transformed and masked satellite image as the query and the original one as the reference. We then train the network to estimate the relative pose between the two satellite images. The training objective is:

$$\mathcal{L}_1 = |\theta - \theta^*| + |t_x - t_x^*| + \left|t_y - t_y^*\right|, \tag{1}$$

where $\theta, t_x$ and $t_y$ denote the network predictions, $\theta^*, t_x^*$ and $t_y^*$ indicate the corresponding ground truth, $\theta$ is the 1-DoF inplane rotation of a ground camera, *i.e.*, the yaw angle, $t_x$ and $t_y$ represent the 2-DoF translation, and $|\cdot|$ denote the $L_1$ norm. After the network has been trained, we substitute the query satellite image with the query ground images during inference for the ground camera's pose estimation, as shown in Fig. 2(b).

The Pose Regressor constructed by neural networks achieves promising rotation estimation performance no matter whether query features are a satellite feature map or a projected overhead-view feature map from a ground view. This is mainly because neural network outputs are inherently sensitive to rotations on the input. A slight rotation difference in input signals results in significant feature differences in deep networks. In contrast, due to aggregation layers such as max-pooling, the high-level deep features inside the neural optimizer may not be sensitive to slight translations on input signals, resulting in poor translation estimation performance. On the other hand, the equivariance property of the convolution operation to translations makes the relative translation between two input signals can be recovered by a spatial correlation. This motivates our network design for translation estimation.

### 3.2 TRANSLATION ESTIMATION BY DEEP METRIC LEARNING

As shown in Fig. 3, a two-branch convolutional network is first applied to the ground and satellite image pair. Each branch is a U-Net architecture and extracts multi-level representations of the original images.

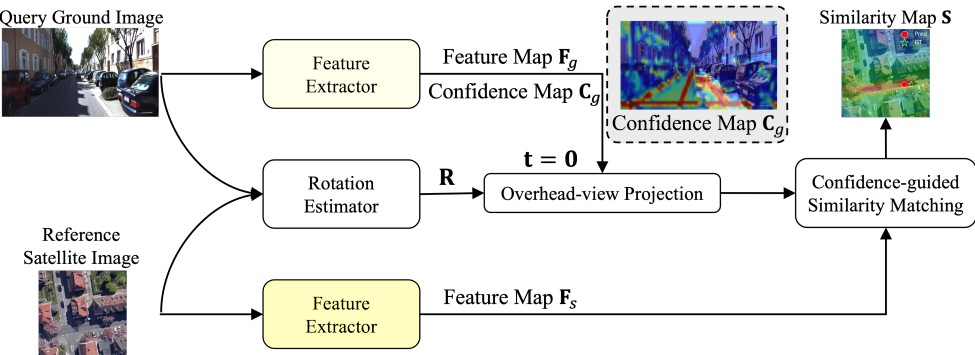

Figure 3: Translation estimation framework. During training, the weights of the rotation estimator are fixed, and we only train the two-branch feature extractors. Other uncolored blocks indicate no trainable parameters are involved.

For the ground branch, not only the feature representation $\mathbf{F}_g \in \mathbb{R}^{H_g \times W_g \times C}$ but also a confidence map $\mathbf{C}_g \in \mathbb{R}^{H_g \times W_g \times 1}$ is extracted. The confidence map is an additional channel of the feature extractor output followed by a sigmoid layer. It indicates whether features at corresponding spatial pixel positions are trustworthy. For example, dynamic objects (*e.g.*, cars) in the images are detrimental to localization performance, while the static road structures are essential features. The higher the confidence, the more reliable the corresponding features. We should note that no explicit supervision is applied to the confidence map. Instead, it is encoded in the cross-view similarity-matching process and learned statistically from the similarity-matching training objective.

We only extract feature representations $\mathbf{F}_s \in \mathbb{R}^{H_s \times W_s \times C}$ for the satellite branch with no confidence map because we empirically found learning confidence maps for satellite images impairs the performance. The reasons hypothesized are twofold: (1) dynamic objects are fewer on the satellite images, and they occupy a relatively smaller region on the satellite image compared to ground-view images; thus, they have a lower impact on localization performance; (2) neither the camera poses, nor the confidence maps, have explicit supervision, thus learning a confidence map for satellite images increases the network training difficulty.

We leverage the trained rotation estimator in Sec. 3.1 to estimate the relative rotation between the ground and satellite images. Then, the ground-view features and confidence maps are projected to the overhead view according to the estimated rotation $\mathbf{R}$ and zero translation $\mathbf{t} = \mathbf{0}$. Similar to that in Fig. 2(b), the overhead-view projection leverages the ground plane Homography. To maintain clarity in the presentation, we utilize the original symbols $\mathbf{F}_g$ and $\mathbf{C}_g$ to represent the projected ground features and confidence maps.

**Confidence-guided similarity matching.** Given that the rotation of the projected overhead-view feature map has been aligned with the observed satellite feature map, the only remaining disparity between them is a translation difference. To compute this translation difference, spatial correlation is utilized. Specifically, the projected overhead-view feature map is used as a sliding window, and its inner product with the reference satellite feature map is computed when aligned at varying locations. This generates a similarity score $\mathbf{S}(u, v)$ between the two view features when aligned at each location $(u, v)$. We provide a visual illustration of this process in the Appendix. The mathematical representation of this spatial correlation process, taking into account ground-view confidence maps, is as follows:

$$\mathbf{S}(u, v) = (\mathbf{F}_s * \widehat{\mathbf{F}_g})(u, v) = \frac{\sum_i \sum_j \mathbf{F}_s(u+i, v+j)\widehat{\mathbf{F}_g}(i, j)}{\sqrt{\sum_i \sum_j \mathbf{F}_s^2(u+i, v+j)}\sqrt{\sum_i \sum_j \widehat{\mathbf{F}_g}^2(i, j)}}, \qquad (2)$$

where $\widehat{\mathbf{F}_g} = \mathbf{C}_g \mathbf{F}_g$ is to highlight important features while suppressing non-reliable features for localization. The pixel coordinate corresponding to the maximum similarity indicates the most likely ground camera's location $(\hat{u}, \hat{v}) = \arg\max_{(u,v)} \mathbf{S}$. This similarity map between the synthesized overhead-view feature map and the reference satellite feature map can also be regarded as the location probability map of the ground camera.

**Supervision.** We apply deep metric learning for network supervision. For a query ground image, we compute a similarity map based on the possible locations between it and its matching and non-matching satellite images, denoted as $\mathbf{S}_{\text{pos}}$ and $\mathbf{S}_{\text{neg}}$, respectively. We maximize the maximum similarity in $\mathbf{S}_{\text{pos}}$ while minimizing the maximum similarity in $\mathbf{S}_{\text{neg}}$:

$$\mathcal{L}_2 = \sum_l log(1 + e^{\alpha(\max \mathbf{S}_{\text{neg}} - \max \mathbf{S}_{\text{pos}})}), \tag{3}$$

where $\alpha$ controls the convergence speed and is set to 10.

When the error of location labels in the training set is the same as the error of locations that we aim to refine during deployment, we only employ Eq. equation 3 for network training. In another scenario where relatively more accurate location labels for ground images in the training data are available than the poses we aim to refine during employment, we introduce an additional training objective to incorporate this signal:

$$\mathcal{L}_3 = \sum_l \left| \max(\mathbf{S}_{\text{pos}}) - \max(\mathbf{S}_{\text{pos}}[u^* - \frac{d}{\gamma} : u^* + \frac{d}{\gamma}, v^* - \frac{d}{\gamma} : v^* + \frac{d}{\gamma}]) \right|, \tag{4}$$

where $(u^*, v^*)$ indicates the location label provided by the training data and has an error of up to $d$ meters, $\gamma$ denotes the ground resolution of the similarity map in terms of meters per pixel. This training objective forces the global maximum in the similarity map to equal a local maximum, with the local region centered at the location label with a radius of $d$ meters.

The whole training objective is:

$$\mathcal{L} = \mathcal{L}_2 + \lambda \mathcal{L}_3, \tag{5}$$

where $\lambda = 0$ indicates such relatively accurate pose labels are unavailable in the training set, while $\lambda = 1$ suggests such labels are available. In our experiments, we set $d = 5$ meters.

### 3.3 OVERALL EVALUATION

After training the networks, the overall evaluation goes through the framework shown in Fig. 3. The input is a query image and its positive satellite image, and the output is an estimated relative rotation between the two images and a location probability map of this query image with respect to this satellite image. The location corresponding to the maximum probability/similarity value is deemed as the query camera location.

## 4 EXPERIMENTS

**Network Architectures.** A UNet-based architecture with a pre-trained VGG16 as the encoder is adopted for feature extraction. The decoder of the UNet is randomly initialized. We empirically found that the feature extractor of satellite images is shareable with ground images captured by a pin-hole camera while not shareable with ground panoramas. This might be because both satellite images and ground images captured by a pin-hole camera map straight lines in the real world to straight lines on images, while panoramas map straight lines in the real world to curves on images. While for different purposes, *i.e.*, rotation and translation estimation, we found non-shareable feature extractors between the two stages help to achieve the best performance. We present detailed analysis and experimental demonstrations in the Appendix. The pose regressor in Fig. 2 is constructed by two swin transformer layers (Liu et al., 2021) followed by two fc layers. This is the same as the neural optimizer architecture in Shi et al. (2023a).

**Dataset.** Our experiments are conducted on a well-known autonomous driving dataset, KITTI (Geiger et al., 2013), and a cross-view localization dataset, VIGOR (Zhu et al., 2021b).

For the KITTI dataset, ground images were captured by a forward-facing pin-hole camera with a limited FoV. The cross-view KITTI dataset includes one training set and two testing sets. Images from Test-1 are from the same region as the training set, while images from Test-2 are from a different region. The location search range for this dataset is around $56 \times 56$ m$^2$, and the orientation noise is $20°$, which follows the official setting as in Shi & Li (2022). The performance evaluation on different initialization errors is presented in the Appendix.

Table 1: Ablation study on the cross-view KITTI dataset. We report the percentage of query images whose locations are restricted to be within $d$ meters of their GT locations along lateral/longitudinal directions and whose orientations are restricted to be within $\theta°$ of their GT orientation, respectively.

| Query | $\mathbf{t}$ Est. | Conf | $\lambda$ | Test-1 (Same-area) | | | | | | Test-2 (Cross-area) | | | | | |
|---|---|---|---|---|---|---|---|---|---|---|---|---|---|---|---|
| | | | | Lateral | | Longitudinal | | Azimuth | | Lateral | | Longitudinal | | Azimuth | |
| | | | | $d=1\uparrow$ | $d=3\uparrow$ | $d=1\uparrow$ | $d=3\uparrow$ | $\theta=1\uparrow$ | $\theta=3\uparrow$ | $d=1\uparrow$ | $d=3\uparrow$ | $d=1\uparrow$ | $d=3\uparrow$ | $\theta=1\uparrow$ | $\theta=3\uparrow$ |
| Sat | Pose Regress. | – | – | 4.88 | 15.13 | 4.80 | 15.64 | 99.89 | 100.00 | 5.06 | 15.46 | 5.25 | 15.79 | 99.95 | 100.00 |
| Grd | Pose Regress. | – | – | 4.88 | 15.11 | 4.74 | 15.66 | 99.66 | 100.00 | 5.04 | 15.46 | 5.29 | 15.83 | 99.99 | 100.00 |
| Grd | Correlation | N | 0 | 45.80 | 78.27 | 6.18 | 16.67 | 99.66 | 100.00 | 45.11 | 73.04 | 6.13 | 18.30 | 99.99 | 100.00 |
| Grd | Correlation | Y | 0 | 59.58 | 85.74 | 11.37 | 31.94 | 99.66 | 100.00 | 62.73 | 86.53 | 9.98 | 29.67 | 99.99 | 100.00 |
| Grd | Correlation | Y | 1 | **66.07** | **94.22** | **16.51** | **49.96** | **99.66** | **100.00** | **64.74** | **86.18** | **11.81** | **34.77** | **99.99** | **100.00** |

The VIGOR dataset contains ground and satellite images from four cities in the US: Chicago, New York, San Francisco, and Seattle. It is divided into same-area and cross-area splits. The same-area split indicates the training and testing images are from the same region (both from the four cities), while the cross-area split adopts images from two cities for training and images from the other two cities for testing. In the original dataset, each ground image has a positive satellite image and several semi-positive satellite images, depending on whether this ground image is within the center $\frac{1}{4}$ region of the satellite image. We follow Lentsch et al. (2023) and use only the positive satellite images.

Since the feature extractors trained for satellite images (Fig. 2(a)) are not applicable to panoramas due to differences in imaging modality, we only evaluate the translation estimation performance on the VIGOR dataset with known and unknown orientations, respectively. The 3-DoF joint location and orientation pose estimation is performed on the KITTI dataset.

**Evaluation Metrics.** We decompose the translation to be along the lateral and longitudinal directions for evaluation on the KITTI and Ford Multi-AV datasets, following the approach of Shi & Li (2022). Specifically, for a query image, we consider it to be successfully localized along a direction if its estimated location along that direction is within $d$ meters of its ground truth (GT) location. Similarly, we consider the rotation estimation to be correct if the estimated rotation is within $\theta°$ of the GT rotation. We record the percentage of successfully localized images along each direction and the percentage of images with correct rotation estimation.

The VIGOR dataset does not provide information on the driving direction of the camera. Therefore, we cannot decompose the translation to be along lateral and longitudinal directions. To evaluate the localization performance on this dataset, we report the median and mean errors of comparison algorithms, following the approach of Xia et al. (2022) and Lentsch et al. (2023).

**Implementation Details.** For training the rotation estimator, we use the positive satellite image for each ground image and randomly transform each image once to create training input pairs. We adopt feature size as $\frac{1}{4}$ of the original image size for both rotation and translation estimation. The original image size is not used because of its large memory consumption in the spatial correlation process in the deep metric-learning training. We use a batch size of $B = 8$ to train the network. In the translation estimation training, each ground image has one matching satellite image and $B - 1$ non-matching satellite images within each batch. Our experiments are conducted on an RTX 3090 GPU. The network is trained for 3 epochs for both stages on the KITTI dataset and 10 epochs on the VIGOR dataset. For the ground images in the KITTI dataset, we use a resolution of $256 \times 1024$. For the VIGOR dataset, we use a resolution of $320 \times 640$. The satellite image resolution is $512 \times 512$ for all datasets. The ground resolution of satellite images is $0.2$ meters per pixel in the KITTI dataset, and $0.111$, $0.113$, $0.118$, and $0.101$ meters per pixel for the cities Chicago, New York, San Francisco, and Seattle in the VIGOR dataset, respectively.

## 4.1 MODEL ANALYSIS

Below, we demonstrate the necessity and effectiveness of each proposed component. The results are presented in Tab. 1.

**(i)** The first row shows the performance of our Pose Regressor (PR) in Fig. 2 with the satellite images as queries (note: not ground images). It can be seen that the estimated rotation of almost all the queries has been restricted to within $1°$ of its GT rotation, while the translation estimation performance is poor, even though there is no domain gap between reference and query images. This

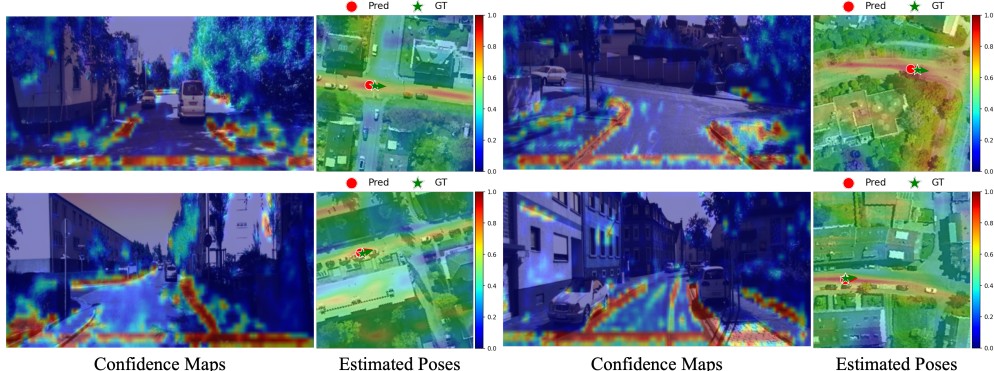

Figure 4: Visualization of learned confidence maps for ground view images and estimated poses by our method.

Table 2: Comparison with the state-of-the-art on the KITTI dataset. ∗ indicates full supervision is adopted.

| Algorithms | Test-1 (Same-area) | | | | | | Test-2 (Cross-area) | | | | | |
|---|---|---|---|---|---|---|---|---|---|---|---|---|
| | Lateral | | Longitudinal | | Azimuth | | Lateral | | Longitudinal | | Azimuth | |
| | $d = 1 \uparrow$ | $d = 3 \uparrow$ | $d = 1 \uparrow$ | $d = 3 \uparrow$ | $\theta = 1 \uparrow$ | $\theta = 3 \uparrow$ | $d = 1 \uparrow$ | $d = 3 \uparrow$ | $d = 1 \uparrow$ | $d = 3 \uparrow$ | $\theta = 1 \uparrow$ | $\theta = 3 \uparrow$ |
| DSM (Shi et al., 2020a)∗ | 10.12 | 30.67 | 4.08 | 12.01 | 3.58 | 13.81 | 10.77 | 31.37 | 3.87 | 11.73 | 3.53 | 14.09 |
| CVR (Zhu et al., 2021b)∗ | 18.61 | 49.06 | 4.29 | 13.01 | - | - | 17.38 | 48.20 | 4.07 | 12.52 | - | - |
| Shi & Li (2022)∗ | 35.54 | 70.77 | 5.22 | 15.88 | 19.64 | 51.76 | 27.82 | 59.79 | 5.75 | 16.36 | 18.42 | 49.72 |
| SliceMatch (Lentsch et al., 2023)∗ | 49.09 | 91.76 | 14.19 | 49.99 | 13.41 | 42.62 | 32.43 | 78.98 | 8.30 | 24.48 | 46.82 | 46.82 |
| OrienterNetSarlin et al. (2023)∗ | - | - | - | - | - | - | 51.26 | 84.77 | 22.39 | 46.79 | 20.41 | 52.24 |
| Shi et al. (2023a)∗ | 76.44 | 96.34 | 23.54 | 50.57 | 99.10 | 100.00 | 57.72 | 86.77 | 14.15 | 34.59 | 98.98 | 100.00 |
| **Ours** ($\lambda = 0$) | 59.58 | 85.74 | 11.37 | 31.94 | 99.66 | 100.00 | 62.73 | 86.53 | 9.98 | 29.67 | 99.99 | 100.00 |
| **Ours** ($\lambda = 1$) | 66.07 | 94.22 | 16.51 | 49.96 | **99.66** | **100.00** | **64.74** | 86.18 | 11.81 | **34.77** | **99.99** | **100.00** |

supports our intuition that a neural network-based regressor's ability to provide accurate translation estimation is limited.

**(ii)** Then, we verify the generalization ability of the Pose Regressor on real ground images. The results in the second row show that the rotation estimation accuracy for real ground images is very close to that for satellite images, demonstrating the effectiveness of our self-supervision training strategy by satellite-and-satellite image pairs.

**(iii)** The third row presents the performance of our method by using spatial correlation for translation estimation and Pose Regressor for rotation estimation. It can be seen that the performance on translation estimation is significantly boosted.

**(iv)** In what follows, we encode the confidence in the spatial correlation process. The results in the fourth row demonstrate the effectiveness of the confidence-guided similarity matching. Fig. 4 provides visualizations of the learned confidence maps and the probability maps of query cameras' location with respect to their matching satellite images. It can be seen that the learned confidence maps are able to ignore dynamic objects and highlight reliable features (e.g., lane lines, road edges).

**(v)** Finally, when relatively accurate (but still noisy) pose labels are available in the training dataset, *i.e.*, the pose error is up to $d = 5$ meters, we set $\lambda = 1$ and use Eq. 4 to incorporate this information in training. The last row of Tab. 1 demonstrates that it successfully boosts the network learning process and improves the performance on both same-area and cross-area evaluation.

## 4.2 COMPARISON WITH THE STATE-OF-THE-ART

In this section, we compare the performance of our method with the recent state-of-the-art, including DSM (Shi et al., 2020a), CVR (Zhu et al., 2021b), Shi & Li (2022), MCC (Xia et al., 2022), SliceMatch (Lentsch et al., 2023), OrienterNet (Sarlin et al., 2023), and Shi et al. (2023a). All these state-of-the-art algorithms adopt full supervision and rely on accurate pose labels for their network, and their results are taken from their original papers or re-evaluated using the author-released models.

Table 3: Comparison with the state-of-the-art on the VIGOR dataset. ∗ indicates full supervision is adopted.

| Algorithms | Same-area | | | | Cross-area | | | |
| | Aligned-orientation | | Unknown-orientation | | Aligned-orientation | | Unknown-orientation | |
| | Mean↓ | Median↓ | Mean↓ | Median↓ | Mean↓ | Median↓ | Mean↓ | Median↓ |
|---|---|---|---|---|---|---|---|---|
| CVR (Zhu et al., 2021b)∗ | 8.99 | 7.81 | – | – | 8.89 | 7.73 | – | – |
| MCC (Xia et al., 2022)∗ | 6.94 | 3.64 | 9.87 | 6.25 | 9.05 | 5.14 | 12.66 | 9.55 |
| SliceMatch (Lentsch et al., 2023)∗ | 5.18 | 2.58 | 8.41 | 5.07 | 5.53 | 2.55 | 8.48 | 5.64 |
| Shi et al. (2023a)∗ | 4.12 | 1.34 | – | – | 5.16 | 1.40 | – | – |
| **Ours** ($\lambda = 0$) | 5.22 | 1.97 | 5.33 | 2.09 | 5.37 | 1.93 | 5.37 | 1.93 |
| **Ours** ($\lambda = 1$) | 4.19 | 1.68 | **4.18** | **1.66** | **4.70** | 1.68 | **4.52** | **1.65** |

The performance comparison between our method and the state-of-the-art on the KITTI dataset is presented in Tab. 2. It can be seen that among all the comparison algorithms, our method achieves the best cross-area evaluation performance, and the performance discrepancy between same-area and cross-area evaluation of our method is the smallest. This is because our method has no information on GT poses, and it is trained to leverage similarity matching for location estimation, preventing itself from overfitting on the GT poses. The performance of our method on the same-area evaluation is slightly inferior to the most recent state-of-the-art, Shi et al. (2023a). However, in the appendix, we demonstrate that the performance of our method on both same-area and cross-area evaluation can be potentially improved when more training image pairs (with pose errors up to tens of meters) are available. Our method does not rely on accurate pose labels of training data. This leads to significant cost and effort savings by eliminating the need for high-precision pose label acquisition.

The performance comparison between our method and the state-of-the-art on the VIGOR dataset is presented in Tab. 3. The training images in the VIGOR dataset are about $4.5\times$ larger than those in the KITTI dataset, and our method achieves comparable performance with Shi et al. (2023a) on same-area evaluation. Furthermore, similar to the observations on the KITTI dataset, the generalization ability of our method from same-area to cross-area is also the best on the VIGOR dataset, with the cross-area evaluation performance surpassing almost all fully-supervised methods.

### 4.3 LIMITATIONS

Although our self-supervised learning approach has achieved promising results, it has a few limitations. (i) First, as explained previously, our self-supervised training strategy for rotation estimation is only suitable for ground images captured by a pin-hole camera. Due to the significant domain differences between panoramic and satellite images, it cannot be applied to estimating a spherical camera's orientation. (ii) Second, our deep metric learning supervision strategy computes the spatial correlation between each query image and several satellite images. To save GPU memory and enable a reasonable batch size for metric learning, we use the feature level of a quarter of the original image size for the translation estimation. This actually sacrifices localization accuracy to some extent. (iii) Finally, similar to all the ground-to-satellite localization networks where a single camera is used for query, our method suffers poor localization performance along the longitudinal direction. This can potentially be addressed using a video or multi-camera setup for the query. We leave these unsolved problems as our future work and encourage the community to pay attention to them.

### 5 CONCLUSION

This paper has introduced the first weakly supervised ground camera pose refinement strategy by ground-to-satellite image registration. Given a coarse location and orientation of a ground camera obtained from noisy sensors or visual retrieval techniques, our method is able to refine this pose by ground-to-satellite image registration using a training dataset without accurate pose labels for ground images. Key components of our approach include a training scheme for ground-and-satellite rotation alignment using satellite-and-satellite image pairs and a deep metric learning supervision mechanism that trains the network for translation estimation. Benefiting from these two innovations, our method, without requiring accurate labels, achieves comparable or superior performance to the recent fully supervised state-of-the-art.

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

# A  DESIGN CHOICE DISCUSSIONS

**Feature extractors.**    We empirically found that the feature extractors for satellite and ground images captured by a pin-hole camera are shareable in the localization task. Tab. 4 presents the comparison of our method with shared or non-shared feature descriptors. In this comparison, the rotation estimator is kept the same and trained only on satellite images. From the results, it can be seen that sharing weights between ground and satellite images achieves better performance. A potential explanation might be that both the satellite images and ground images captured by a pin-hole camera map straight lines in the real world to straight lines in images and the viewpoint differences of the two view images are solved by a geometry projection module. This is similar to the task of multi-view stereo and image-based rendering, where the feature extractors for multi-view images are shared, and their differences are handled by Homography/geometry warping. Not sharing weights between the two branches increases the learning burden of the network, especially when supervision is not strong, resulting in inferior performance.

While for rotation and translation estimation, we found different feature extractors for different purposes achieve better performance. Tab. 5 illustrates the comparison results. This might be because good features for rotation and translation estimation are not identical. When re-using the feature extractors in rotation estimation for translation, we found the network converges slowly, and the performance on both rotation and translation estimation is poor, although better than the original coarse poses that we aim to refine.

Table 4: Sharing feature extractors or not between satellite and ground images captured by a pin-hole camera.

| | Share? | Test-1 (Same-area) | | | | | | Test-2 (Cross-area) | | | | | |
| | | Lateral | | Longitudinal | | Azimuth | | Lateral | | Longitudinal | | Azimuth | |
| | | $d=1\uparrow$ | $d=3\uparrow$ | $d=1\uparrow$ | $d=3\uparrow$ | $\theta=1\uparrow$ | $\theta=3\uparrow$ | $d=1\uparrow$ | $d=3\uparrow$ | $d=1\uparrow$ | $d=3\uparrow$ | $\theta=1\uparrow$ | $\theta=3\uparrow$ |
| **Ours** ($\lambda=0$) No | No | 48.64 | 77.37 | 9.97 | 25.63 | 99.66 | 100.00 | 54.49 | 79.75 | 8.96 | 26.54 | 99.99 | 100.00 |
| **Ours** ($\lambda=0$) Yes | Yes | 59.58 | 85.74 | 11.37 | 31.94 | 99.66 | 100.00 | 62.73 | 86.53 | 9.98 | 29.67 | 99.99 | 100.00 |
| **Ours** ($\lambda=1$) No | No | 62.81 | 93.00 | 19.90 | 55.53 | 99.66 | 100.00 | 62.81 | 84.77 | 13.14 | 36.89 | 99.99 | 100.00 |
| **Ours** ($\lambda=1$) Yes | Yes | 66.07 | 94.22 | 16.51 | 49.96 | 99.66 | 100.00 | 64.74 | 86.18 | 11.81 | 34.77 | 99.99 | 100.00 |

Table 5: Sharing feature extractors or not for rotation and translation estimation.

| | Share? | Test-1 (Same-area) | | | | | | Test-2 (Cross-area) | | | | | |
| | | Lateral | | Longitudinal | | Azimuth | | Lateral | | Longitudinal | | Azimuth | |
| | | $d=1\uparrow$ | $d=3\uparrow$ | $d=1\uparrow$ | $d=3\uparrow$ | $\theta=1\uparrow$ | $\theta=3\uparrow$ | $d=1\uparrow$ | $d=3\uparrow$ | $d=1\uparrow$ | $d=3\uparrow$ | $\theta=1\uparrow$ | $\theta=3\uparrow$ |
| **Ours** ($\lambda=0$) | Yes | 33.77 | 74.66 | 9.17 | 26.13 | 11.69 | 34.64 | 32.18 | 71.92 | 7.58 | 24.00 | 12.64 | 37.24 |
| **Ours** ($\lambda=0$) | No | 59.58 | 85.74 | 11.37 | 31.94 | 99.66 | 100.00 | 62.73 | 86.53 | 9.98 | 29.67 | 99.99 | 100.00 |
| **Ours** ($\lambda=1$) | Yes | 35.62 | 81.69 | 10.68 | 31.78 | 10.20 | 31.35 | 32.62 | 73.72 | 8.87 | 26.07 | 10.21 | 31.62 |
| **Ours** ($\lambda=1$) | No | 66.07 | 94.22 | 16.51 | 49.96 | 99.66 | 100.00 | 64.74 | 86.18 | 11.81 | 34.77 | 99.99 | 100.00 |

**Overhead-view projection.**    This paper leverages deep metric learning for network supervision. It is well-recognized that a small batch size in metric learning affects performance negatively. Moreover, Harley et al. (2023) also confirms that a larger batch size impacts more significantly the overall performance than different overhead-view projection methods. Thus, this paper adopts a simple method that leverages ground plane homography for overhead view projection. It consumes the least GPU memory among the existing BEV synthesis methods and thus enables a large batch size. We achieved a batch size of 8 on an RTX 3090 GPU with this simple projection method.

We tried leveraging the geometry-guided cross-view transformer in Shi et al. (2023a) for the overhead view projection. However, we can only achieve a batch of 4 on the same GPU, and the performance is inferior to the simple ground plane Homography-based overhead view projection, as demonstrated in Tab. 6. We leave the balance between a better overhead-view feature synthesis module and less GPU memory consumption as future work.

# B  VISUALIZATION OF PROJECTED OVERHEAD-VIEW FEATURE MAPS

We provide some examples of the projected overhead-view feature maps, the reference satellite feature maps, and the estimated poses by our method in Fig. 5. For the projected overhead-view feature map in Fig. 5 (b), its center corresponds to the query camera location, and the right direction

Table 6: Comparison between different overview synthesis methods. The method of "Transformer" is from Shi et al. (2023a).

| | | Test-1 (Same-area) | | | | | | Test-2 (Cross-area) | | | | | |
| | | Lateral | | Longitudinal | | Azimuth | | Lateral | | Longitudinal | | Azimuth | |
| | | $d=1\uparrow$ | $d=3\uparrow$ | $d=1\uparrow$ | $d=3\uparrow$ | $\theta=1\uparrow$ | $\theta=3\uparrow$ | $d=1\uparrow$ | $d=3\uparrow$ | $d=1\uparrow$ | $d=3\uparrow$ | $\theta=1\uparrow$ | $\theta=3\uparrow$ |
|---|---|---|---|---|---|---|---|---|---|---|---|---|---|
| **Ours ($\lambda=0$)** | Transformer | 48.74 | 83.14 | 8.27 | 24.20 | 99.66 | 100.00 | 52.69 | 79.99 | 8.46 | 24.00 | 99.99 | 100.00 |
| | **Homography** | 59.58 | 85.74 | 11.37 | 31.94 | 99.66 | 100.00 | 62.73 | 86.53 | 9.98 | 29.67 | 99.99 | 100.00 |
| **Ours ($\lambda=1$)** | Transformer | 62.02 | 92.13 | 16.01 | 46.54 | 99.66 | 100.00 | 62.97 | 87.54 | 11.36 | 33.70 | 99.99 | 100.00 |
| | **Homography** | 66.07 | 94.22 | 16.51 | 49.96 | 99.66 | 100.00 | 64.74 | 86.18 | 11.81 | 34.77 | 99.99 | 100.00 |

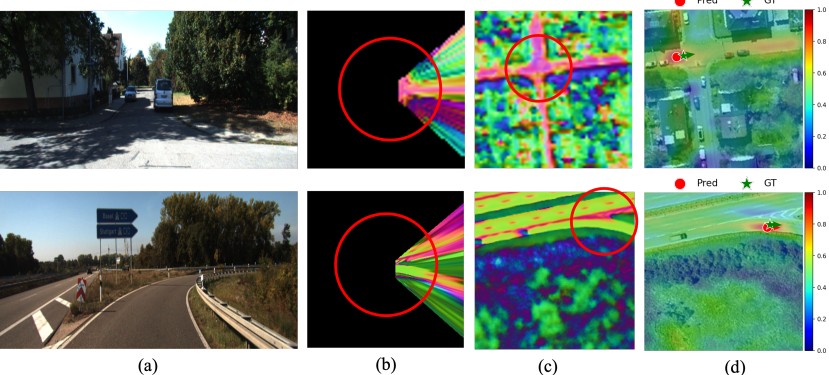

Figure 5: (a) Ground image; (b) Synthesized overhead view feature map from the ground image; (c) Satellite image feature map; (d) Estimated location probability map and camera orientation indicated by the red arrow.

corresponds to the camera heading. The positive satellite image's feature map is presented in Fig. 5 (c). The coverage of Fig. 5(b) and Fig. 5(c) is the same in this visualization, which is around $100 \times 100$ m$^2$. Our Pose Regressor compares the projected overhead-view feature map and the reference satellite feature map and predicts a relative orientation between them, as shown by the red arrow in Fig. 5 (d). We then rotate the projected feature map in Fig. 5 (b) according to the estimated relative orientation and align its orientation with the reference satellite feature map, Fig. 5 (c). A spatial correlation is next conducted between them to estimate the location probability map of the ground camera with respect to the satellite image. This process is illustrated next.

## C  VISUAL EXPLANATION OF THE SPATIAL CORRELATION PROCESS

The spatial correlation process is illiustrated in Fig. 6. We first center crop the synthesized overhead view feature map depicted in Fig. 5 (b) and make its coverage around $40 \times 40$ m$^2$, as we consider scene contents within 20m to the camera location is the most important for the localization purpose. Then, we adopt the cropped overhead view feature map as the correlation kernel, similar to the yellow kernel in Fig. 6, and the reference satellite image as the correlation input, indicated by the green grid in Fig. 6, and apply inner product between the input and the kernal. The output, indicated by the pink grid in Fig. 6, is the location probability map of the ground camera with respect to the satellite image.

In practice, the coverage of the reference satellite map and the kernel is engineered to make the coverage of the convolution output slightly larger than the location search space of the ground camera. In this example, the coverage of the satellite map is around $100 \times 100$ m$^2$, and that of the convolution output (location probability map) is about $60 \times 60$ m$^2$.

**Comparison between projecting features and images.**    In this paper, we follow the general practice of projecting features instead of images (Shi et al., 2023b). This is because when projecting ground images to an overhead view by assuming ground plane Homography, the pixels for scene objects above the ground plane are incorrectly projected to the overhead view and thus suffer distortion. In this way, the scene information of these objects will be lost in the projected image, resulting in inferior localization performance.

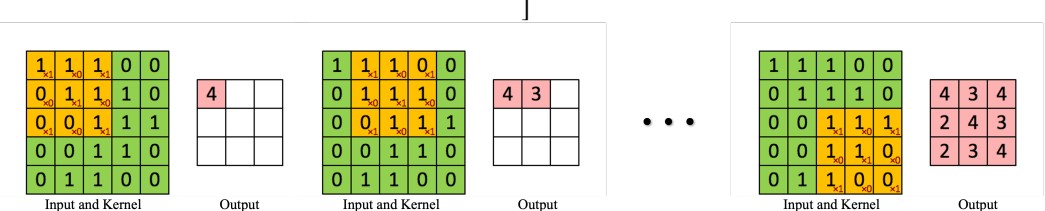

Figure 6: The spatial correlation process. We compute the inner product between the reference satellite feature map (input) and synthesized overhead view feature map (kernel) from the ground image across all possible locations. This figure is from `https://giphy.com/gifs/blog-daniel-keypoints-i4NjAwytgIRDW`

Table 7: Comparison between projecting features and images.

| | | Test-1 (Same-area) | | | | | | Test-2 (Cross-area) | | | | | |
| | | Lateral | | Longitudinal | | Azimuth | | Lateral | | Longitudinal | | Azimuth | |
| | | $d=1\uparrow$ | $d=3\uparrow$ | $d=1\uparrow$ | $d=3\uparrow$ | $\theta=1\uparrow$ | $\theta=3\uparrow$ | $d=1\uparrow$ | $d=3\uparrow$ | $d=1\uparrow$ | $d=3\uparrow$ | $\theta=1\uparrow$ | $\theta=3\uparrow$ |
|---|---|---|---|---|---|---|---|---|---|---|---|---|---|
| **Ours** ($\lambda=0$) | Images | 45.40 | 80.57 | 6.97 | 21.39 | 99.92 | 100.00 | 48.24 | 79.16 | 6.95 | 20.60 | 100.00 | 100.00 |
| | **Features** | 59.58 | 85.74 | 11.37 | 31.94 | 99.66 | 100.00 | 62.73 | 86.53 | 9.98 | 29.67 | 99.99 | 100.00 |
| **Ours** ($\lambda=1$) | Images | 54.31 | 90.59 | 13.36 | 38.32 | 99.92 | 100.00 | 58.53 | 87.34 | 11.24 | 33.23 | 100.00 | 100.00 |
| | **Features** | 66.07 | 94.22 | 16.51 | 49.96 | 99.66 | 100.00 | 64.74 | 86.18 | 11.81 | 34.77 | 99.99 | 100.00 |

In contrast, features have a larger field of view of the original image and encode higher-level semantic information about the scene. For example, the building roots also have a semantic meaning of "building". It can be mapped to the building roof in the overhead view, which shares the same semantic information as the building root. Thus, projecting features instead of the original images can tolerate the errors in the overhead-view projection by the ground plane Homography to some extent. We illustrate the experimental comparison between projecting features and images in Tab. 7. Not surprisingly, projecting features achieves better performance.

# D    SENSITIVENESS TO INITIALIZATION ERRORS

In this section, we investigate the performance of our method under varying pose initialization errors.

**Range of location errors.** Table 8 presents the performance comparison between our method and the state-of-the-art, Shi & Li (2022) and Shi et al. (2023a), across different ranges of location errors: $28 \times 28$ m$^2$ and $56 \times 56$ m$^2$, while maintaining the same orientation ambiguity of $20°$. The results show that our method achieves consistently the best performance on cross-area evaluation.

**Variation in orientation ambiguity.** Subsequently, we augment the orientation ambiguity from $20°$ to $80°$, while maintaining a location error range of $56 \times 56$ m$^2$. Table 9 provides the performance comparison between our method, Shi & Li (2022) and Shi et al. (2023a). Our method achieves the best cross-area evaluation performance on the different orientation ambiguity. Furthermore, the results reveal a decline for all methods in the percentage of images for which the estimated orientation is restricted to $1°$ as the orientation ambiguity increases. Nevertheless, our method and Shi et al. (2023a), which is recently accepted to ICCV2023, consistently maintain the majority of image orientations within a $3°$ margin from their ground truth values. Consequently, the translation estimation performance remains robust. In contrast, Shi & Li (2022) encounter a notable drop in both translation and orientation estimation performance.

# E    MODEL SIZE AND EVALUATION SPEED COMPARISON

We present the model size and evaluation speed comparison with two recent state-of-the-art, whose models and evaluation scripts have been released, in Tab. 10. All of them are evaluated on an RTX 3090 GPU. It can be seen that our method achieves the fastest evaluation speed with a relatively small model size.

Table 8: Performance comparison with different location error ranges on the cross-view KITTI dataset.

| Location Error Range | Algorithms | Test-1 (Same-area) | | | | | | Test-2 (Cross-area) | | | | | |
|---|---|---|---|---|---|---|---|---|---|---|---|---|---|
| | | Lateral | | Longitudinal | | Azimuth | | Lateral | | Longitudinal | | Azimuth | |
| | | $d=1\uparrow$ | $d=3\uparrow$ | $d=1\uparrow$ | $d=3\uparrow$ | $\theta=1\uparrow$ | $\theta=3\uparrow$ | $d=1\uparrow$ | $d=3\uparrow$ | $d=1\uparrow$ | $d=3\uparrow$ | $\theta=1\uparrow$ | $\theta=3\uparrow$ |
| $28\times28$ m² | Shi & Li (2022)* | 44.66 | 73.92 | 12.06 | 35.62 | 25.31 | 57.41 | 34.17 | 72.30 | 11.56 | 35.08 | 11.40 | 48.18 |
| | Shi et al. (2023a)* | 85.85 | 98.46 | 23.27 | 46.99 | 98.89 | 99.97 | 60.01 | 87.96 | 14.69 | 35.64 | 99.42 | 100.00 |
| | **Ours** ($\lambda=0$) | 61.46 | 87.76 | 13.44 | 38.14 | 99.76 | 100.00 | 65.62 | 90.32 | 13.46 | 38.53 | 99.97 | 100.00 |
| | **Ours** ($\lambda=1$) | 66.39 | 94.38 | 18.18 | 53.59 | 99.76 | 100.00 | 67.90 | 89.76 | 14.29 | 42.92 | 99.97 | 100.00 |
| $56\times56$ m² | Shi & Li (2022)* | 35.54 | 70.77 | 5.22 | 15.88 | 19.64 | 51.76 | 27.82 | 59.79 | 5.75 | 16.36 | 18.42 | 49.72 |
| | Shi et al. (2023a)* | 76.44 | 96.34 | 23.54 | 50.57 | 99.10 | 100.00 | 57.72 | 86.77 | 14.15 | 34.59 | 98.98 | 100.00 |
| | **Ours** ($\lambda=0$) | 59.58 | 85.74 | 11.37 | 31.94 | 99.66 | 100.00 | 62.73 | 86.53 | 9.98 | 29.67 | 99.99 | 100.00 |
| | **Ours** ($\lambda=1$) | 66.07 | 94.22 | 16.51 | 49.96 | 99.66 | 100.00 | 64.74 | 86.18 | 11.81 | 34.77 | 99.99 | 100.00 |

Table 9: Performance comparison with different location error ranges on the cross-view KITTI dataset.

| Orientation Ambiguity | Algorithms | Test-1 (Same-area) | | | | | | Test-2 (Cross-area) | | | | | |
|---|---|---|---|---|---|---|---|---|---|---|---|---|---|
| | | Lateral | | Longitudinal | | Azimuth | | Lateral | | Longitudinal | | Azimuth | |
| | | $d=1\uparrow$ | $d=3\uparrow$ | $d=1\uparrow$ | $d=3\uparrow$ | $\theta=1\uparrow$ | $\theta=3\uparrow$ | $d=1\uparrow$ | $d=3\uparrow$ | $d=1\uparrow$ | $d=3\uparrow$ | $\theta=1\uparrow$ | $\theta=3\uparrow$ |
| $20°$ | Shi & Li (2022)* | 35.54 | 70.77 | 5.22 | 15.88 | 19.64 | 51.76 | 27.82 | 59.79 | 5.75 | 16.36 | 18.42 | 49.72 |
| | Shi et al. (2023a)* | 76.44 | 96.34 | 23.54 | 50.57 | 99.10 | 100.00 | 57.72 | 86.77 | 14.15 | 34.59 | 98.98 | 100.00 |
| | **Ours** ($\lambda=0$) | 59.58 | 85.74 | 11.37 | 31.94 | 99.66 | 100.00 | 62.73 | 86.53 | 9.98 | 29.67 | 99.99 | 100.00 |
| | **Ours** ($\lambda=1$) | 66.07 | 94.22 | 16.51 | 49.96 | 99.66 | 100.00 | 64.74 | 86.18 | 11.81 | 34.77 | 99.99 | 100.00 |
| $80°$ | Shi & Li (2022)* | 26.95 | 62.39 | 5.14 | 15.69 | 3.10 | 8.88 | 22.43 | 54.63 | 5.17 | 15.78 | 3.05 | 8.50 |
| | Shi et al. (2023a)* | 70.21 | 95.47 | 22.29 | 48.90 | 53.27 | 93.98 | 56.97 | 87.72 | 15.17 | 35.39 | 58.68 | 95.92 |
| | **Ours** ($\lambda=0$) | 53.11 | 86.03 | 12.99 | 32.18 | 57.65 | 96.79 | 57.68 | 84.92 | 11.64 | 31.52 | 56.79 | 97.76 |
| | **Ours** ($\lambda=1$) | 57.94 | 91.49 | 17.73 | 47.44 | 57.65 | 96.79 | 60.50 | 86.57 | 12.62 | 35.60 | 56.79 | 97.76 |

Table 10: Model size and evaluation speed comparison on the KITTI dataset.

| Model Size | | | Evaluation Speed | | |
|---|---|---|---|---|---|
| Shi & Li (2022) | Shi et al. (2023a) | **Ours** | Shi & Li (2022) | Shi et al. (2023a) | **Ours** |
| 20.2 M | 29.1 M | 20.6 M | 500 ms | 200 ms | 47 ms |

# F  PERFORMANCE WITH DIFFERENT AMOUNTS OF DATA AS SUPERVISION

Below, we analyze the performance of our method with $\lambda=0,1$ and the state-of-the-art, Shi & Li (2022) and Shi et al. (2023a), when different amounts of training data are employed. The results are illustrated in Fig. 7.

For most models, except Shi & Li (2022), we observe a consistent increase in performance on the same-area evaluation (Test-1) as the amount of training data increases. However, when it comes to the cross-area evaluation (Test-2), the two state-of-the-art methods, which require ground truth pose for supervision, exhibit a decline in performance when the training data exceeds $80\%$. This phenomenon suggests that our method avoids overfitting and holds the potential for further performance improvements with additional training data. Moreover, it's worth noting that our method doesn't necessitate GT labels for ground images during training, simplifying the process of large-scale data collection and reducing associated costs.

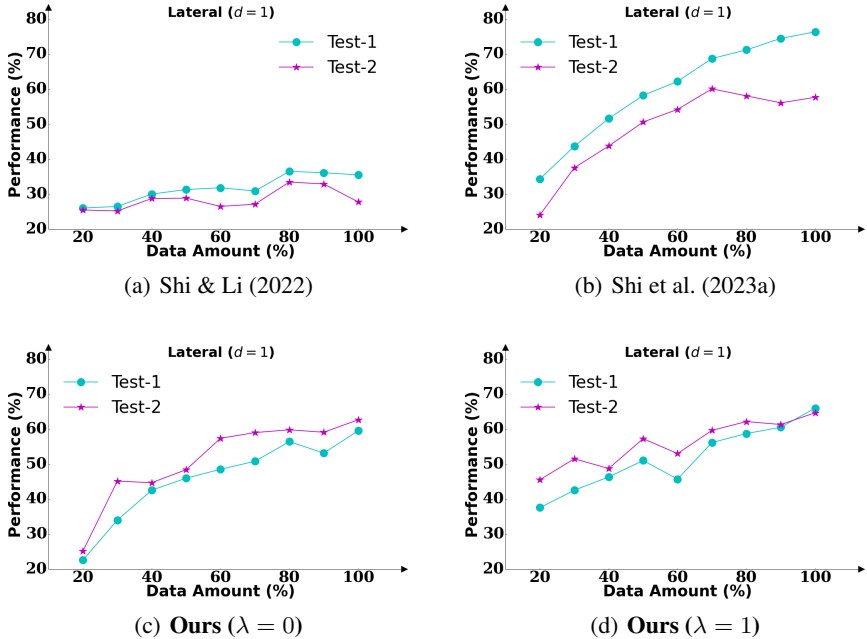

Figure 7: Performance comparison between our method and the state-of-the-art on the KITTI dataset with different amounts of training data.

