# OpenReview forum: "Weakly-supervised Camera Localization by Ground-to-satellite Image Registration"
_ICLR.cc/2024/Conference — Submitted to ICLR 2024_

### Official Review · Reviewer_db9o · 2023-10-29

**Soundness:** 3 good
**Presentation:** 3 good
**Contribution:** 3 good
**Rating:** 6
**Confidence:** 4

**Summary:**

This paper focuses on camera localization with ground-to-satellite registration but uses a weakly supervised setting. It assumes that the pose label during training is not accurate. The proposed method first estimates the orientation with a self-supervision. Then the translation is estimated using feature similarity matching which is supervised by a soft-margin triplet loss. The proposed method is evaluated on KITTI and VIGOR datasets with quantitative comparisons. It outperforms most of the SOTA methods, except a recent work in ICCV 2023.

**Strengths:**

+ The writing is mostly clear.
+ The motivation and problem formulation are interesting.
+ The proposed method is evaluated on two major datasets with different settings. Although it does not outperform some SOTA methods, the result is still impressive for a weakly supervised method.
+ The proposed method is more robust on initialization errors than previous methods.
+ The limitations and other discussions are also very interesting.
+ The visualizations and figures are well presented.

**Weaknesses:**

-	My main concern is about the setting of the proposed method. Given that the accurate pose label is not available for some places, it is still possible to get small-scale accurate labels at known locations. In this case, a semi-supervised setting could be more suitable than a weakly supervised formulation. The small-scale accurate label could also help improve the performance if properly designed.
-	Some suggestions to think about (Not a weakness): It might also be interesting to consider larger errors for the training images, for example, the satellite-view image has a certain probability of being an incorrect match of the query but is still close to the ground-truth locations.
-	There is still room for improvement in the writing. The motivation part of using the proposed approach could be improved:” Leveraging a blackbox network to regress……”. Minor issues: RTK is not explained when it first appears.

**Questions:**

-	One advantage of weakly supervised learning is that more training data could be used. There might be some ways to demonstrate that more training data could be obtained with less accuracy requirement on the label.

---

> ### Author Response · Authors · 2023-11-17
>
> We thank Reviewer-db9o for the valuable comments. Please find our response below, and let us know your further questions.
>
> **1. problem setting.**
>
> Thank you for suggesting the semi-supervised application. Our method can be easily adapted to address the scenario when a small amount of training data with accurate pose labels is available by adding additional supervision to the network with this amount of data.
>
> Below, we show the performance of our method when fine-tuned with different portions of the current training dataset with accurate pose labels. Data amount = 0 indicates the original weakly supervised pipeline where no such data is available.
>
>
> |Data Amount (%)|0|     0.2|0.3     |0.4|0.5|0.6|0.7|0.8|0.9|1|
> |-|-|-|-|-|-|-|-|-|-|-|
> |Lateral ($d=1$)|62.73|39.49|44.61|55.31|60.96|65.60|71.48|74.61|78.61|83.86|
>
> The results show that fine-tuning leads to better performance when the data amount with accurate pose labels is over 50%. However, when the data amount is smaller, the fine-tuning leads to inferior performance. We suspect this is because the model overfits the limited training data with accurate labels, causing a loss of general inference ability on other data.
>
> We agree a properly designed training method should address this problem, and we would like to explore this further in our future work.
>
> Apart from this semi-supervised scenario, we believe the original weakly-supervised application addressed in this paper is also of high importance. It avoids additional effort to select the portion of accurate data, as most localization algorithms and noisy GPS sensors themselves do not provide a measure of whether its prediction is accurate or not.
>
> **2. Consider larger errors for the training images.**
>
> We are sorry that we did not really understand the question. Let us try our best to explain something according to our understanding, and please let us know your further questions.
>
> When the location error of training images is larger, we can increase the coverage of the corresponding satellite image to make it still cover the query image's local surroundings. Our framework is still applicable.
>
> When the satellite image is an incorrect match of the query (the actual location of the camera is outside the region covered by the satellite image, and this is not known), our method still compares the similarity between the ground and satellite image across locations on the satellite image, and find the location corresponds to the largest similarity (though the largest similarity is very small).
>
> When the satellite image covers the local surroundings of the ground camera, while due to time differences between the two-view images, the ground structure might be changed or the illumination is significantly different (day and night), our method behaves similarly as above. It still compares the similarity between the two view images across possible locations on the satellite image and finds the location corresponds to the largest (though very small) similarity.
>
> **3. The motivation part could be improved.**
>
> Thanks for the suggestion. We have rewritten the introduction to make the motivation clearer. The revised content is marked as blue.
>
> **4. RTK is not explained when it first appears.**
>
> We have added the full description of RTK in the abstract: Real Time Kinematics (RTK)
>
> **5. How to obtain more training data with less accuracy requirement on the label?**
>
> A consumer-level mobile phone with built-in GPS, or a consumer-level noisy GPS equipped in a vehicle with a camera, can help to obtain more training data.
>
> The papers below suggest that GPS-enabled smartphones are typically accurate to **within a 4.9m radius** under open sky and **within around 20m** in dense high rise environments. The experiments conducted in this paper set the GPS error to be within a 56m x 56m region (a 28m radius). Thus, it is easy to use consumer-level smartphones to collect more data for network training.
>
> [1]. Menard T, Miller J, Mowak M, Norris D. Comparing the GPS capabilities of the Samsung Galaxy S, Motorola Droid X, and the Apple iPhone for vehicle tracking using FreeSim_Mobile. 14th International IEEE Conference of Intelligent Transportation Systems. 2011, Oct. Washington, DC.
> [2]. Mok E, Retcher G, Wen C. Initial test on the use of GPS and sensor data on modern smartphones for vehicle tracking in dense high rise environments. Proceedings of the Ubiquitous Positioning, Indoor Navigation, and Location Based Services 2012. 2012, Oct. Helsinki, Finland.

---

> > ### Comment · Reviewer_db9o · 2023-12-04
> >
> > Thanks for the clarification in the rebuttal. It addresses most of my concerns and I will keep my original rating.

---

### Official Review · Reviewer_mQu3 · 2023-10-31

**Soundness:** 3 good
**Presentation:** 3 good
**Contribution:** 3 good
**Rating:** 6
**Confidence:** 4

**Summary:**

The paper proposes a technique able to register satellite images with images taken from ground, e.g. by a camera mounted on a car. The results are promising on existing datasets, especially when compared to other existing methods. The idea setting this technique apart from others is the abandonment of rigorous exploitation of ground truth camera poses. The paper is well structured and reads easily. Estimating rotation and translation of ground cameras on large scale is a significant problem in many applications. The paper, altough not explicitely mentioned, target representation learning as it addresses the gap in feature representation between overhead satellite images and perspective
 images taken by the ground camera.

**Strengths:**

Registration of large baseline images is still a challenging problem, especially when it comes to the extreme. Registration of top view and ground images is especially difficult as image features for matching are rare, e.g. because of occlusion, out of plane rotation, etc. The idea in this work is to circumvent this matching problem by considering CNN features on much higher level with much larger receptive fields and semantic information.
Another idea of the technique is the possible avoidance of ground truth camera poses needed for training the model. The inference of rotation is thereby trained by a self-supervised learning approach, while translation is interpreted as correlation problem between satellite images and ground images. The training is supervised by a few labeled ground truth poses similar to metric learning problems.
The results in table 1 are promising, altough a gap between the solutions w vs. w/o knowledge of ground truth poses still exists. The comparison with other SOTA methods clearly show superiority of the technique (table 2).

**Weaknesses:**

The paper is well structured and readable, however I suggest to improve the introduction by motivating in much more clarity the matching problem and the idea of how you represent the problem of learning the high-level features and the registration via the correlation. The approach seems to be analog to popular representation learning approaches using contrastive learning. By focusing in you introduction and motivation on these problems and novelties, the paper would make clear its contributions to representation learning, the main topic of this conference.

At some places in the paper I found incorrect statements:
- page 2: "neural networks are inherently sensitive to rotations ..." in its generality this statement needs to be refused as e.g. equivariant networks address this problem (https://arxiv.org/pdf/2205.07362.pdf)
- page 2: "... synthesised overhead view image ..." the sentence gives the reader the impression that satellite images are always "overhead view" orthographic images which of course is not the case
- page 4: "magnitude of the rotation" ?
- page 4: "the equivariance property of convolution to translations ..." convolution is invariant to translation
- page 5: Eq 2 descibes the correlation coefficient but later on page 14 a convolution was used

**Questions:**

- page 4: Eq.1. the rotation is parametrised so why not writing this? R(a)
- page 4: what does |R-R*| mean? L1 of ... this formulation needs a more precise writing
- would it not be benificial to minimise R^(-1)*R → I ?, as R is SO(3)

The satellite images used in this work are 512x512pixel. In practice satellite images are orders of magnitude larger. To be useful in practice, the method needs to find regions of interest in satellite images in order to perform registration. How could you achieve this?

---

> ### Author Response · Authors · 2023-11-17
>
> We thank Reviewer-mQu3 for the valuable comments. Please find our response below, and let us know your further questions.
>
> **Suggestion: Improve the introduction and make contributions clear to representation learning.**
>
> Thank you very much for the suggestion. We have rewritten the introduction part and labeled the revised text as blue.
>
>
> **Incorrect statements:**
>
> **1: "Neural networks are inherently sensitive to rotations." Equivariant networks address this problem.**
>
> This description is avoided after we rewrite the introduction.
>
> **2: "synthesized overhead view image" gives the reader the impression that satellite images are always "overhead view" orthographic images, which of course is not the case**
>
> Thank you for the suggestion. We have added a footnote in this paper, clarifying that the satellite images used *for ground camera localization* are roughly orthographic.
>
>
> **3: "magnitude of the rotation" ?**
>
> We have changed this description to "magnitude of the rotation angle".
>
> **4: "the equivariance property of convolution to translations ..."**
>
> We have changed the description to "the equivariance property of the convolution operation to translations".
>
>
> **5: Eq 2 describes the correlation coefficient, but later on page 14, a convolution was used**
>
> We have changed the terms to make them consistent.
>
>
> **Questions:**
>
> **1: Parameterization of \mathbf{R} in Eq. 1 and Eq. 1 needs more precise writing.**
>
> Thanks for the suggestion. We have changed Eq.1 to use the original angle representation instead of $\mathbf{R}$ in the loss. The revised content is marked as blue.
>
> **2: Would it not be benificial to minimise $\mathbf{R}^{-1}\mathbf{R}$ → $\mathbf{I}$ ?, as $\mathbf{R}$ is SO(3)**
>
> Thanks for the suggestion. We make our network directly output the angle instead of a rotation matrix and thus do not need to apply this constraint to the network output.
>
>
> **3: Satellite image resolution is 512 x 512. What if satellite images are orders of magnitude larger?**
>
> In this work, we consider refining a coarse location of a ground image by ground-to-satellite registration. Thus, given this coarse location and its statistical error, we can extract a satellite image that covers the local surroundings of the query image. In this work, the satellite image coverage is around 100m x 100m (with a resolution of 512 x 512).
>
> When the location ambiguity of a ground image is significantly large, for example, at the city or state level, cross-view retrieval is usually applied. We discuss this line of work in the first paragraph of the Related Work section.

---

> > ### Comment · Reviewer_mQu3 · 2023-12-05
> >
> > Thanks for your response. I ask the authors to improve on three essentials:
> >
> > 1) In your introduction where you connect your idea with representation learning, you need to add references. E.g. in the sentence: "We resort to the recent representation learning approaches by contrastive learning to solve this problem.".
> > To make the idea more clear, I suggest to adapt Fig. 2a or add a new figure to make the training even more clear by interpreting Fig. 2 in https://arxiv.org/pdf/2002.05709.pdf for you approach
> >
> > 2) Orthographic projection is an assumption of your method, unless you show evidence that your method can also handle projective transformations. Please, write the text in this way.
> >
> > 3) In method: the maths is not clear enough; the rotation angle is theta and you should write rotation angle theta and R(theta) respectively.
> >
> > If these corrections are done well, I recommend to accept the paper.

---

### Official Review · Reviewer_9PqU · 2023-11-02

**Soundness:** 3 good
**Presentation:** 3 good
**Contribution:** 2 fair
**Rating:** 3
**Confidence:** 5

**Summary:**

This paper proposes to register an image captured on the ground to a satellite image in two stages. In the first stage, the ground image is projected to a bird's-eye view, and the 2D rotation and translation are estimated between the projected ground image and the satellite image. In the second stage, the 2D translation within the rotated ground image is re-estimated by dense feature patch matching. To facilitate the training, the authors synthesize bird's-eye view images with known rotation and translation as self-supervision from the satellite images. To demonstrate the effectiveness of the proposed methods, the authors conduct experiments on the KITTI and Vigor datasets and achieve better performance than some recent works on ground-satellite image registration.

**Strengths:**

- The method is clearly introduced and easy to implement.
- Self-supervision from the bird's-eye view synthesis is a reasonable design.
- The experiments are well conducted in terms of the evaluated metrics and ablation studies.

**Weaknesses:**

In general, the paper lacks sufficient insights and contributions:
- The bird's-eye view synthesis in Sec. 3 needs to be further investigated and justified.
   - The synthesized bird's-eye view could be dramatically different from the projected bird's-eye view from the ground image due to parallax, occlusion, and appearance changes. It is very difficult to learn feature representations that are invariant to these changes.
   - In order to convince the audience that the gap is small, the authors should a) introduce their data augmentation in detail, b) analyze the feature differences between the projected and actual bird's-eye view images.
- An important reference is missing: "OrienterNet: Visual Localization in 2D Public Maps with Neural Matching", CVPR '23.
  - The proposed method in this paper is very similar to this missing reference except for the self-supervision. The OrienterNet has achieved better performance than the paper, and more importantly, it is better written with deeper insights.
  - To highlight the difference, the authors should clearly describe why precise supervision is difficult, which does not seem to be the case for me since we can get precise reconstructions even from a mobile phone and then register a sequence of images to the satellite images.

**Questions:**

To decouple the contribution between model structure and supervision, can we do supervised training and see the upper-bound?

---

> ### Author Response · Authors · 2023-11-17
>
> We thank Reviewer-9PqU for the valuable comments. Please find our response below, and let us know your further questions.
>
> **1: More Justification on the BEV synthesis in Sec. 3.**
>
> Our method does not aim to learn viewpoint invariant features robust to parallax and occlusion. Instead, we want to explore the co-visible regions between the two views and weaken other regions' effect in similarity matching. The co-visible regions mainly concentrate on the ground plane and thus are handled by the ground plane's homography projection. For regions that are not co-visible, we resort to the confidence map to lower their weights in similarity matching. The appearance discrepancy for the co-visible areas is handled by the end-to-end training.
>
> We discussed why using this simple projection instead of a powerful yet complex cross-view transformer for overhead-view synthesis in Sec. A of the appendix. The main reason is a balance between batch size and the model complexity. A large batch size is important in self/weakly supervised learning by contrastive learning. We also visualized the projected and corresponding satellite feature maps in Sec. B. The ground structures co-visible by both views are aligned well. More discussions are presented in Sec. C.
>
> We follow previous works for data augmentation: none is applied to ground images, while satellite images are randomly rotated.
>
> **2: Similar to a missing reference OrienterNet (CVPR23)?**
>
> Thank you for pointing out this accidentally missed reference. We have added it back to the paper.
>
> Both works synthesize an overhead-view feature map from the ground-view image and match it against the reference satellite feature map for translation estimation. The **differences** are:
> 1. OrienterNet rotates the synthesized overhead view feature map K times and then matches them to the reference satellite image. To avoid the additional search space caused by rotation and obtain continuous rotation estimates, we present a pose regressor for rotation estimation.
> 2. Our pose regressor leverages the sensitivity of a normal neural network to the rotation of input signals, and we introduce a self-supervised method for this pose regressor training. Our rotation estimation performance is significantly better than OrienterNet.
> 3. Our method does not require ground truth location labels of ground images for network training.
>
> **3: Why precise supervision is difficult? Precise reconstruction can be done from a mobile phone and then register a sequence of images to the satellite images.**
>
> Precise reconstruction can be obtained from a sequence of ground view images by SfM. However, the GPS of a mobile phone is often noisy. Thus, the absolute geo-location of the reconstructed 3D structure is unknown. Registering reconstructed 3D structures to a satellite image requires complex engineering techniques. From our investigation, there is no readily available technique for this. This effort can be avoided by using the method proposed in this paper. The original noisy GPS signals of ground images can be directly used for network training.
>
> **4: Inferior performance to OrienterNet?**
>
> OrienterNet reports its performance on Test-2 of the KITTI dataset with a single query image, and the model is trained on the original training dataset of KITTI (the same setting as ours). The comparison results are as follows:
> || Lateral|| Longitudinal|| Orientation||
> |-|-|-|-|-|-|-|
> ||$d=1$|$d=3$|$d=1$|$d=3$|$\theta=1^\circ$|$\theta=3^\circ$|
> |OrienterNet|51.26|84.77|22.39|46.79|20.41|52.24|
> |Ours($\lambda=0$)|62.73|86.53|9.98|29.67|99.66|100.00|
> |Ours($\lambda=1$)|64.74|86.18|11.81|34.77|99.99|100.00|
>
> The results of OrienterNet are from Tab.3 of its original paper (OrienterNet(c)).
>
> Except for the longitudinal pose estimation performance, which is inherently ambiguous in ground-to-satellite image matching, our method achieves significantly better lateral pose and orientation estimation than OrienterNet, even though OrienterNet leverages the ground truth poses for training. In contrast, our method does not have access to this.
>
> **5: Better written with deeper insights.**
>
> Thank you for the suggestion. We have revised our introduction, marked as blue, to convey this paper's idea better.
>
> **6: Performance with supervised training.**
>
> The performance of our method with full supervision is as follows:
> ||Lateral||Longitudinal||Orientation||
> |-|-|-|-|-|-|-|
> ||$d=1$|$d=3$|$d=1$|$d=3$|$\theta=1^\circ$|$\theta=3^\circ$|
> |Test-1|83.86|96.69|30.74|58.65|99.66|100.00|
> |Test-2|50.99|79.51|14.62|31.25|99.99|100.00|
>
> The performance improves significantly for the same-area evaluation (Test-1), while it suffers slight degradation on the cross-area evaluation (Test-2). This is similar to the observation on other fully-supervised methods: models tend to overfit on the training area when ground truth poses are available and further demonstrates that the proposed weakly-supervised approach exhibits strong generalization ability.

---

### Author Response · Authors · 2023-11-20

Dear Reviewers,

We greatly appreciate your valuable comments on this paper.
As the discussion phase is nearing its conclusion, may we kindly ask whether we have adequately addressed your questions?

Should you have any further questions or require additional clarification, please do not hesitate to let us know.

Best regards,

The Authors

---

### Meta-Review · Area_Chair_hnUs · 2023-12-15

**Metareview:**

This paper proposes a technique to register satellite images with ground-to-satellite images in two stages. The first stage involves projection of the ground image to a bird's-eye view, estimating 2D rotation and translation between the two images. The second stage involves re-estimating the translation within the rotated ground image using dense feature patch matching. The authors synthesize bird's-eye view images with known rotation and translation from satellite images. The method outperforms most ground-to-satellite image registration methods, except a recent work in ICCV 2023. The paper addresses the gap in feature representation between overhead satellite images and perspective images taken by the ground camera.

## Strengths

• The method is easy to implement and self-supervised from a bird's-eye view synthesis.
• Experiments are conducted with evaluated metrics and ablation studies.
• The technique circumvents the matching problem by considering CNN features at a higher level with larger receptive fields and semantic information.
• The technique also avoids ground truth camera poses needed for training the model.
• The training is supervised by labeled ground truth poses similar to metric learning problems.
• Results are promising, but a gap exists between solutions with and without ground truth poses.
• The method is more robust on initialization errors than previous methods

## Weaknesses

• The bird's-eye view synthesis in Sec. 3 needs further investigation and justification.
• Learning feature representations invariant to parallax, occlusion, and appearance changes is challenging.
• The authors should introduce their data augmentation and analyze feature differences between the projected and actual bird's-eye view images.

Lack of Reference: "OrienterNet: Visual Localization in 2D Public Maps with Neural Matching", CVPR '23.
• The proposed method in this paper is similar to the missing reference, except for self-supervision.
• The authors should clearly explain why precise supervision is difficult.

Suggestions for Improvement:
• Clarify the matching problem and the problem of learning high-level features and registration via correlation.
• Consider a semi-supervised setting for the proposed method.
• Consider larger errors for the training images.
• Improve the motivation part of using the proposed approach.
• Explain RTK when it first appears.

**Justification For Why Not Higher Score:**

The paper should include all the reviewer comments to improve before being accepted.

**Justification For Why Not Lower Score:**

N/A

---

### Decision · Program_Chairs · 2024-01-16

Reject